PERSPECTIVE

# Recommendations for achieving interoperable and shareable medical data in the USA

Ana Szarfman [1✉], Jonathan G. Levine [2], Joseph M. Tonning [3],
Frank Weichold [1], John C. Bloom [4], Janice M. Soreth[5],
Mark Geanacopoulos[1], Lawrence Callahan [1], Matthew Spotnitz [6], Qin Ryan[1],
Meg Pease-Fye [1], John S. Brownstein [7], W. Ed Hammond [8],
Christian Reich [9] & Russ B. Altman [10]

## Abstract

Easy access to large quantities of accurate health data is required to understand medical and scientific information in real-time; evaluate public health measures before, during, and after times of crisis; and prevent medical errors. Introducing a system in the USA that allows for efficient access to such health data and ensures auditability of data facts, while avoiding data silos, will require fundamental changes in current practices. Here, we recommend the implementation of standardized data collection and transmission systems, universal identifiers for individual patients and end users, a reference standard infrastructure to support calibration and integration of laboratory results from equivalent tests, and modernized working practices. Requiring comprehensive and binding standards, rather than incentivizing voluntary and often piecemeal efforts for data exchange, will allow us to achieve the analytical information environment that patients need.

Reported world-wide mortality from COVID-19 has surpassed 6 million with over 16% of deaths in the USA alone[1]. Despite our vaccination efforts against COVID-19, the analytical deficiencies of USA health information systems (HIS) uncovered by the pandemic remain largely unresolved[2]. We still cannot answer basic questions that should be answerable by a simple query of the data, such as, what is the mortality rate according to patient variables? Also, public health systems and practitioners are still forced to rely on outmoded forms of communication (e.g., paper and fax) which do not provide rapid access to needed information.

Although recognized as a leader in advancing cutting edge biomedical research and medical technology, the USA continues to rely on multiple, independent healthcare systems and versions that cannot seamlessly communicate with each other. This lack of interoperability within and across hospital systems, laboratories, public health programs, physicians' offices, and regulatory and research data resources hinders rapid improvements in medical treatment, public health, decision-making, and research. The main reason for the failure to achieve interoperability and for the information loss, inefficient operations, and huge (and frequently hidden) costs that result, is the lack of comprehensive, centrally coordinated, fully validated, traceable, and enforceable medical data collection and transmission standards[3]. Bi- and

[1] U.S. Food and Drug Administration, Silver Spring, MD 20993, USA. [2] Independent Researcher, Rockville, MD 20855, USA. [3] Your Health Concierge, Inc., Bethesda, MD 20815, USA. [4] College of Veterinary Medicine, Purdue University, West Lafayette, IN 47907, USA. [5] Independent Medical Consultant, Chevy Chase 20815 MD, USA. [6] Columbia University Irving Medical Center, New York, NY 10032, USA. [7] Harvard Medical School, Boston Children's Hospital, Boston, MA 02115, USA. [8] Duke Center for Health Informatics, Duke Clinical & Translational Science Institute, Durham, NC 27705, USA. [9] Columbia University Department of Biomedical Informatics, New York, NY 10032, USA. [10] Departments of Bioengineering, Genetics & Medicine, Stanford University, Stanford University School of Medicine, Stanford, CA 94305-4145, USA. ✉email: ana.szarfman@fda.hhs.gov

multi-directional feedback loops that are needed for prompt access to ancillary data, for clarifications, and for quickly reporting and addressing system and data errors are also lacking. Without easy access to this additional information, electronic health records (EHRs) cannot be made portable[4], and the full potential of these records to support research and innovation cannot be realized.

Data that can be easily exchanged is a goal that many parties have long been advocating for. The COVID-19 pandemic has made this issue more urgent than before as we have *"to move faster than the virus"* [Personal communication from Dr. Mirta Roses]. Unfortunately, to date the COVID-19 pandemic has only underscored the consequences of having information systems that rely on non-binding standards for data management and exchange, standards that are themselves based on multiple, unreconciled data models. In principle, a data model should provide universal definitions of data elements (i.e., units of data having a precise meaning and interpretation) for users of heterogenous data sites that want to share or aggregate data, to allow them to speak a common language[5].

Emerging technologies that promise to revolutionize healthcare add additional urgency to efforts to achieve interoperability in our HIS. In a decade, there will be more sources of data[6], for example from wearable devices such as the Apple Watch and Fitbit, that patients will use to record information. By combining these data with artificial intelligence and machine learning, in which machines are able to automatically process the data, diagnosis and prediction of patient outcomes could be improved. However, the successful use of these computational tools strongly depends on accurate collection and exchange of massive amounts of complex data derived from next-generation sequencing, imaging devices, laboratory assays, and many other sources. Unfortunately, the data being collected remain predominantly in local silos and frequently the data are neither standardized nor of the quality required by these advanced automated approaches[7,8]. The promise of all of these technological and scientific advances will be unrealized without interoperable standards that are fully representative of real-world clinical data (not just based on theoretical examples), and are fit-for purpose for data collection, exchange, integration, and analysis, and traceable to the original information.

Perhaps the most challenging roadblock for implementing interoperability for data collection is the tolerance for highly customized, proprietary HIS and their unique versions. The inconsistencies created by unnecessary customization creates a state of confusion that makes it impossible to reliably identify in a timely fashion critical data facts and inconsistencies that must be communicated to those making critical decisions about how these systems should be designed and implemented. These decision makers include those in government organizations, the vendors of HIS, software developers, and other stakeholders, including patients and patient advocates.

By presenting the following description of deficiencies in the USA health information system and recommendations for addressing them at their root causes, we hope to stimulate constructive dialog among multiple stakeholders and inform policy changes in the USA and other countries where such measures are needed. Recognizing our ethical responsibility to rapidly provide the best information to help patients[9], we propose building an alternative, more transparent system based on interoperability that starts at the data collection stage. This alternative system would be one in which the benefits of new computational technologies can be realized, where patients are able to take control of their data, and where accurate and timely data can be rapidly shared to advance medical research and improve public health.

## The lack of universal and harmonized data collection and transmission standards

To date, policies to increase interoperability in our HIS have been based on downstream transactions, for example they seek to improve e-prescribing, billing, health information exchange, certification of EHRs[10,11], and regulatory submissions. These policies do not enforce a universal standard for the collection and transmission of defined variables and values for each data element, even for straightforward information such as demographic data[5]. Lacking universal standards, most health data exchange is therefore subject to the custom constraints of a multitude of unique, proprietary HIS, and the non-interoperable, disparate versions of the data elements in these systems. For example, proprietary HIS systems modify most lab data received in their databases by mapping them to built-in terms. This results in a multitude of data conversion cycles that are difficult to document and untangle, because they are not traceable to the original data elements. Mapping and remapping from the irregular internal codes of each HIS version to the standardized versions needed for exchanging data is an error-prone, inefficient, and costly process that is repeated in reverse at the receiving end(s) when integrating the exchanged data back into the internal codes of the HIS version in which they were received.

There are ongoing efforts by the Office of the National Coordinator for Health Information Technology and the Centers for Medicare & Medicaid Services in the U.S. to support data exchange via secure FHIR HL7 application programming interfaces[12] (i.e., software exchange engines created by The Health Level Seven International healthcare standards organization). However, without common, enforceable, and well-documented data structure and coding across pertinent HIS and application programming interfaces, data exchange may still require manual, and frequently blinded mapping, which makes rapid transfer of information unfeasible.

To achieve a health system that enables continuous improvements, we need systems that collect the data that are most important for patient care, for accomplishing critical analyses, for enhancing the level of evidence, and for addressing public health challenges[13–16]. Therefore, we must focus on developing universal standards for the collection and validation of the most clinically important data as they are created (e.g., results from centrally calibrated laboratory tests during the entire course of clinical care). Only when such standards are in place can we ensure that valid information is being correctly captured and delivered. We must also ensure that the diverse software, transfer engines, and information technology systems can correctly interpret these standards, and process standard nomenclatures and notations without corruption. Redundant backup systems, feedback loops for prompt and early identification and communication of problems, and automated data verification processes will be needed to ensure data integrity and identify and correct the sources of transmission errors. Options should be provided for the public to monitor the accuracy of their medical data throughout all encounters (e.g., prescriptions, diagnosis, procedures), in the same way they can monitor their interactions with the Social Security System or banking institutions.

A recent collaborative effort between HL7 International, which provides common standards for exchange of data in healthcare, and the Observational Health Data Sciences and Informatics (OHDSI) collaborative, which defines and maintains the common data model known as OMOP for international observational research studies, seeks to implement a unique data model for assembling and sharing information gathered in clinical care. This undertaking should enable us to integrate clinical data within huge repositories for advanced analytics, without the information loss caused by sequential mapping and remapping

from and to a multitude of untraceable data models[17]. However, HL7 and OHDSI are not providing interoperable standards for the collection of factual data into EHRs. Without strong legislative support, funding, and enforcement, an interoperable model for data collection at the source that can fully address our most critical health information needs will not become a reality.

Although recommendations for addressing data interoperability in our HIS are described in the policy documents of organizations involved in oversight[11] and in the scientific literature[18], much of the medical and scientific community remains insufficiently aware of the limitations of these systems, and tackling our widespread usability problems has not become a universally shared priority. We suggest that recent failures in disease prediction models[19–21] can be attributed in part to irregularities in how data are captured, exchanged, and maintained, and our inability to systematically access and compare these data across multiple EHR systems and versions over time.

To build quality data systems, we must have reliable enforcement mechanisms in place to monitor the implementation of and adherence to interoperable data standards. To monitor such process, we need to conduct Good Clinical Practice inspections and adopt reliable monitoring tools and enforcement mechanisms (analogous to those used by the Treasury Department to assure honesty of monetary transactions). These inspections will require highly trained professionals capable of detecting inaccurate data, improper coding, and failures of prediction models that clinicians rely on.

There are limits to the time healthcare professionals can (and should) spend entering data. A core principle of informatics is that data should only be entered once, and whenever possible by the device collecting the data. In the scenarios where automated entry is not possible, an interoperable system should facilitate data entry and coding by providing automated, interactive graphic representations of the data already in the system and smarter options with standard terminology for outcomes for given symptoms, diseases, medications, and patient profiles. Establishing high quality hardware and software systems for collecting and delivering interoperable and fully traceable healthcare data to users would also create a dynamic in which it would be easier to assess the value and cost of data, and what additional data should be captured. Furthermore, the creation and support with reimbursable billing codes of large numbers of positions for scientifically trained clinical information professionals to manage medical information systems will increase the value of these systems for caregivers, researchers, and patients.

## Issues requiring prompt attention

**Lack of ascertainment of unique patients**. Although HIPAA initially required the creation of a health identifier in 1996[22], federal funds for unique universal patient identifiers have been banned since Congress prohibited their use due to privacy concerns[23]. Our failure to implement national, unique identifiers in the USA linking a patient's data to their healthcare professionals and HIS systems leads to unlinked, incomplete, and often duplicated records, and is another significant source of data quality problems that have been avoided in countries that have implemente unique identifiers[23]. In addition, it is still nearly impossible for a person to access their own vaccination records if they are in databases separate from their EHR records or were submitted by paper or fax. It is also difficult or impossible to carry out the early cancer prevention studies[24] that require that complete clinical information be linked to the correct patients even when they change health providers.

Although the prospect of unique patient identifiers raises valid privacy concerns, it can be argued that it would be easier to monitor and protect privacy with a single, properly encoded universal identifier than with a multitude of poorly documented ones. The absence of a unique identifier is actually one of the greatest causes of invasion of privacy, because typically over half of the EHRs in an institution will mistakenly include someone else's data [Personal communication by Dr. W. Ed Hammond] that may be identifiable.

The current reliance on data aggregation techniques to protect patient privacy significantly delays our access to the information and impedes our understanding of the trajectory of diseases in individual patients, with potentially adverse consequences for their medical care and for identifying critical patient-level variables for subsequent research studies. We must therefore invest in better and updated privacy protection systems and law enforcement solutions. As data scientists, we are concerned about the limitations of HIPAA for privacy protection, due to the ease that such data can be re-identified. Our laws and regulations need to balance individual privacy protection, with making data available for improving health outcomes. At a minimum, the approach to governance we adopt must ensure the following: the system is able to identify and control who can have the authorized level of access to the medical records; every user has a unique ID and a secure password; audit trails are used to track every user activity, and to provide accountability; only authorized personnel can access audit trails, and assess who has accessed or modified a record; and the data storage provider is not able to access personal identifiable information.

A single patient identifier also has health equity ramifications in its favor. Patients who are poorer typically have less insurance coverage or none at all and often switch healthcare systems. They are underrepresented in HIS and research studies, and less likely to have their specific needs understood. A unique identifier should improve the representation of these patients in our HIS and thus our ability to address health inequities.

**Lack of information about patient mortality**. The inadequacy of our current system for data collection is well illustrated by our failure to collect data as fundamental as mortality in a standardized fashion. Fatal outcomes are not incorporated into the medical record unless death occurs during hospitalization. When needed for public health measures, epidemiological studies, and other research, data on death may be obtained from private services that collect information from funeral homes and obituaries, disease registries unconnected to EHRs, or from the National Death Index website. This website is typically late in gathering mortality information as it is collected by a multitude of disparate local and state systems before being reported to the National Center for Health Statistics. Comprehensive data on mortality and cause of death should be methodically linked to clinical data for the over 330 million individuals in the USA (as we have begun to do for COVID-19 cases). This information will allow for the creation of focused decision support systems for clinical data that are better designed to prevent serious and fatal medical errors, one of the top causes of death in hospitals in the USA[25].

**Poorly codified and calibrated clinical laboratory data**. Clinical laboratories began collecting digitized data in the 1960s. Although these data support 60 to 70 percent of decisions related to diagnosis, treatment, hospital admission, and discharge, they remain poorly codified, complicated to process, and are underused for medical decision-making and research.

USA programs that defined the minimum government standards for EHRs have offered laboratories incentives to adopt proposed standards for messaging and encoding laboratory data.

Unfortunately, serious functional problems still exist with the coding of laboratory test identifiers. There are multiple ways for the same analytes to be represented by different labs and instruments and this results in improper assessments of coded terms and incorrect code selection and categorization. Moreover, coding systems often do not allow for transparent incorporation and transmission of the limits of detection of a test, the presence of interfering substances, and how a particular analyte is measured. Also, failure to enforce the use of consistent quantitative units of measure is a frequent source of data errors.

There is a pressing need for an expanded infrastructure to support the collection and distribution of the stable reference standards needed to support the accurate calibration and safe integration of the results from equivalent tests measuring the same analyte, performed by different instrument platforms or laboratories[26,27]. The Office of the National Coordinator for Health Information Technology recognizes this problem when it states, "Harmonization status indicates calibration equivalencies of tests and is required to verify clinical interoperability of results. Tests that are harmonized may be interpreted and trended together, and may use the same calculations, decision support rules, and machine learning models. Tests that are not harmonized should be interpreted and processed individually, not in aggregate with other tests."[3]

This infrastructure will simplify the identification of a natural functional interoperability pathway that can be used as a backbone for integrating the currently unwieldy, inconsistent, and incomplete data coding standards for laboratory data. An illustration of the consequences of the failure to fully standardize laboratory data collection and calibration of the results is the limited understanding of the evolving prevalence of COVID-19, due to the inability to account for the performance differences of the over 1,000 SARS-CoV-2 diagnostics that are listed worldwide[28]. We also need to understand their performance characteristics according to the particular purpose for which a test is being performed (e.g., permission to travel, to access specific facilities, etc.)[29].

**Business practices that hinder modernization**. The world-wide-web and online business transaction systems such as Amazon's e-commerce system were built with a clear understanding of the value of interoperability. These systems ensure that the correct data are collected and stored in an organized, automatically aligned format that is optimized to address new communication requirements and analytical functions. Realizing this scenario for health data will require changes in current practices. Since individual enterprises have built one-of-a-kind systems, there are often strong financial reasons not to share proprietary information. Current laws prohibiting information blocking have not accomplished their purpose, because it is impossible to effectively oversee the thousands of unique versions of HIS.

Given this scenario, it would be useful, once the needed information and data routes are identified and categorized, to develop prototype systems to demonstrate the benefits of profound change in how we manage health information. The development, testing, and validation of these prototypes for addressing the various requirements of patient care and research and development should be based on the integrity, completeness, traceability, and usability of the data; on the avoidance of preventable medical errors; and on measurable improvements in health outcomes.

**Improving the processing of laboratory data linked to the regulatory activities of the FDA**. In contrast to other data transactions for which federal regulations are seeking to increase interoperability (e.g., using ICD-10 coding for billing), in the USA there is no clear business model that incentivizes standardization of laboratory data coding and its integration across medical encounters. Nor is there a single coordinated authority in the USA to monitor and enforce the adoption of, and adherence to, such standards or the transmission of intact laboratory data to end users. Interoperable standards for laboratory data are still very immature (paper and fax lab submissions are still commonplace), and still rely on billing codes for managing and understanding this information, despite their limited scope. For example, there are only 12 Current Procedural Terminology codes used for billing reimbursement that identify the COVID-19 or SARS-COV-2 infectious agent or their antibody response[30], while the FDA lists 357 identifiers for COVID-19 testing [31].

Therefore, we suggest that one area that we should use as a model for how to achieve interoperability of patient data, and where favorable incentives for reform may already exist, is in the processing of clinical laboratory data in drug marketing applications submitted to the FDA. Currently, such data undergo multiple transformation steps before regulatory submission, and although results in a given new drug application may be calibrated, the results for many equivalent analytes coming from different sponsors, laboratories, and instruments are not necessarily calibrated the same way[3,26,27].

We propose to begin the process of prototype development by creating a centralized calibration process for routine and critical analytes so that results collected during clinical trials will be equivalent regardless of the instrument or the laboratory. The aim is to eliminate the severe problems that result from customized data systems and demonstrate that time-consuming mapping and translation errors, and the associated loss of information, can be avoided while adding traceability and clarity to the clinical laboratory data in marketing applications. The recent phenomenon of increased mergers between central labs supporting pharmaceutical company sponsors and labs that support hospital networks will enable the systematic identification and removal of many deficiencies that derived from multiple sources of lab data, and help implementation of robust and universal data standards. We expect that the time and cost savings and the gains in accuracy demonstrated by a prototype system for clinical laboratory data will be welcomed by the pharmaceutical and device industries, the research and public health communities, and patients. In its processing of lab data, this initiative will include all the standardized data elements needed for analysis of regulatory data submissions, including those related to demographics, diagnosis, medical history, laboratory tests, death, and cause of death. Such standards will greatly enhance regulatory review of marketing applications across multiple sponsors and facilitate comparison of clinical trial lab results across applications, providing valuable feedback to the pharma sponsors.

When it reaches a level of maturity, the prototype for handling laboratory and other clinical data in regulatory submissions could be expanded to non-regulatory contexts, including routine patient care. The lessons learned could eventually be applied to the evaluation and certification of EHRs and decision support systems. The knowledge gained in how to create a truly interoperable system could also be used to address the analytical needs of other data resources including registries, repositories of real-world data, and regional data exchanges.

**Returns on investment**

Adoption of our recommendations will simplify the continual enhancement, maintenance, oversight and the analytical

---

**Box 1: ▎ Recommended steps for legislative action**

- Empower an oversight and enforcement agency with a qualified advisory board representing all stakeholders to identify and address critical usability requirements for building an interoperable & interconnected Health Information System in the USA
- Enforce the creation and maintenance of a thorough common data model for clinical data
- Prohibit unwieldy data customization by enforcing interoperable and interconnected standards for medical data collection for every organization that collects or processes medical data (e.g., hospitals, laboratory information systems)
- Establish automated data verification processes to confirm that the data collected are transmitted without distortion to correct patient records and end users; identify problems through feedback loops, and correct the sources of any data errors
- Enforce a standard, certifiable calibration process that ensures that different tests for a given analyte give equivalent results regardless of the instrument used or the laboratory performing the test
- Implement a universal, unique patient identifier secured by the strongest privacy-enhancing technology and supported by a security infrastructure
- Authorize a central body to collect death and cause of death information for all individuals in the USA, with the federal government defining the requirements and precautions needed to avoid fraud
- Require the Centers for Medicare & Medicaid Services to create reimbursable billing codes for clinical informatics professionals who can make informed decisions about HIS selection, optimization of analytical and decision support functions, and maintenance
- Establish Good Clinical Practices with adequate inspections of the analytical clinical data processes and facilities
- Create incentives aligned with patients' and public health needs in which healthcare vendors are rewarded for documenting and avoiding medical errors and unnecessary processing costs
- Identify and correct gaps and inconsistencies in current regulatory requirements

---

**Box 2: ▎ Recommended steps for public-private partnership action**

- Catalog the information required for building the decision support systems needed to improve the quality of patient care, and maintain updated information
- Authorize an advisory board representing all stakeholders, subject matter experts, and patient advocates to identify and address current roadblocks for collecting and transmitting interoperable clinical data in a fully traceable manner
- Work with Standards Development Organizations to determine how best to codify clinical data at the point of data collection, and maintain continuous quality improvements in coding
- Identify the shortest pathways for information to reach the correct patient record and authorized end users
- Adopt data collection systems that are fully traceable to the original data facts and thus make it possible to locate the sources of medical and system errors to avoid their recurrence
- Identify and link important health information data that are currently not connected to hospitals, including death, cause of death, data collected by registries, and vaccinations
- Implement a pilot system for collection of standardized, calibrated clinical laboratory data, and ancillary information, starting with the clinical data submitted to the FDA by drug companies
- Assess progress based on measurable improvements in data integrity and completeness, analytics, healthcare delivery, and auditability, as well as reduced operating costs
- Make the data elements and documentation of the mature prototype(s) available in a public repository for testing and feedback from users

---

functions of a fully interoperable, public health and medical data system. The savings achieved through interoperability across the Research and Development value chain would expedite the discovery and development of safe and effective vaccines, treatments, and the identification of marketed drugs that can be repurposed to treat patients. Analysts will be able to discover consistent and reproducible efficacy and safety signals within and across multiple data resources and perform meta-analyses of all selected data rather than focusing on limited, static summary reports[32,33]. Automated analytical tools will remain securely linked to the original data, making it possible to quickly complete additional evaluations of emerging issues. In support of these predictions, countries that have more interconnected HIS have been able to analyze their medical data more efficiently, and have provided important findings. For example, the rapid completion of the dexamethasone study in the United Kingdom in patients with COVID-19[34,35] would have been very difficult to achieve in the USA with our highly customized and uncoordinated systems for capturing patient-level data. Also, Israel with its standardized, highly interoperable medical information system and a universal patient identifier has provided critical information about

breakthrough infections in patients who were considered to be fully vaccinated [36,37].

In a fully interoperable health information system, patients will receive improved medical care based on the ability of clinicians to detect, troubleshoot, and prevent critical and costly medical and system errors and benefit from public health measures that are based on information that is reliable, complete, and up-to-date. Although the deconstruction and rebuilding that we are proposing may be costly, it will be even more expensive to continue to undertake never-ending customization and processing of data to fit the unpredictable, continuously changing constraints of multiple, incompatible silos. To generate popular support for transforming our HIS, we must inform the public of the risks of medical errors associated with a failure to accurately transmit critical information to the correct patient record and the risks for data quality posed by the current methods of maintaining data confidentially and privacy. Once we achieve true interoperability, it will become obvious that all health data are important, and we are ethically bound to adopt data retention policies that will preserve this information for our needs and for future generations.

## Conclusions

In conclusion, the COVID-19 crisis is another wake-up call that reminds us that we cannot continue to use outdated data solutions that jeopardize our ability to advance research capabilities[4], and can lead to medical errors and loss of life. Boxes 1 and 2 offer a set of summary recommendations that, if adopted, would help achieve needed solutions to the problems being described in this paper.

If Amazon can track packages, international banks can track money, and weather maps can track complex weather patterns, we can also learn how to track and analyze complex health data. Our recommendations are intended to create the conditions in which we can address an entrenched and highly complex problem that will only become worse if unaddressed. This problem will not be cheap to fix, but it will be much costlier to ignore.

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

## Acknowledgements

The views expressed in this article are those of the authors and do not necessarily represent the views or policies of the Food and Drug Administration or of the other institutions. The authors wish to acknowledge the valuable insights and discussions with Norman Stockbridge, Robert Temple, Mitra Rocca, Helena Sviglin, Frank Pucino, and Gregory Pappas from the U.S. Food and Drug Administration; Ingeborg Holt, informatician; Andrea Pitkus, Laboratory Informaticist and Clinical Terminology expert; Riki Merrick, Association of Public Laboratories; Sharona Hoffman, Case Western Reserve University School of Law; Nanguneri Nirmala, Director, Center for Clinical Evidence Synthesis, Tufts Medical Center; Mirta Roses Periago, World Health Organization Special Envoy on COVID-19 for Latin America and the Caribbean and Sir George Alleyne, both Directors Emeritus of the Pan American Health Organization; and Sean Khozin, Chief Executive Officer, CancerLinQ and former Associate Director of the FDA Oncology Center of Excellence.

## Author contributions

A.S., J.G.L., J.M.T., F.W., J.C.B., J.M.S., M.G., L.C., M.P. designed the study, developed critical concepts, and wrote the paper; C.R., W.E.H., J.C.B., R.B.A. added clarity to critical concepts; M.S., Q.R. contributed to the work methodology; all authors read, edited, and approved the final version of the paper.

## Competing interests

The authors declare no competing interests.
