## [Peer Review File · Communications Medicine]

Reviewers' comments:

Reviewer #1 (Remarks to the Author):

— Summary of paper —

This publication is a summary of recommendations to improve and modernize healthcare systems towards a world where an increasing range of software is used, where patients are increasingly taking control of their data and where sharing data for public health purposes with multiple public and private organizations is increasingly important.

At its heart the paper recommends a shift to standards for data collection. This is an important goal, and recommendations for healthcare organizations are helpful.

— Summary of review —

This is an important set of recommendations and I would recommend publication, but some points could be improved / expanded.

— Main comments —

There are a few points that the introduction could mention that these recommendations may help address:

The introduction does not mention some of the main emerging healthcare trends that necessitate solutions to the challenges the paper outlines, which I think is a missed opportunity. In 10 years time, there will be many personal health records and apps (from EMR providers, tech companies, government agencies etc) that patients will use to store health data. If healthcare providers are not able to share data with these, patients may choose to go elsewhere. Patients often use apps to track their pregnancy or health, often sharing updates on social media. It is not hard to think that clinics which can interact with such apps will be more popular with patients - and given the public health benefits of PHRs (some governments internationally are promoting them), in years to come healthcare systems may be mandated to store patient generated data from wearables like Apple Watch / fitbit.

There is also no mention of AI, which again relies on well curated datasets. As more organizations start to see benefits from AI in medical imaging, chatbots, etc, hospitals will miss out on the benefits without interoperability and data collection standards.

Finally I wonder if genomics will play a larger part in treatment, particularly in cancer treatment. Even at a local level, I think the ability to store this type of data is very poor, and certainly from a research perspective the scientific community will miss out on the opportunity for large scale GWAS studies without a better standard for data collection.

The paper outlines 5 major problems (no standards for data collection, no 'national PID', unreliable mortality data, poor clinical coding, EMR systems that are incentivized to lock a healthcare org into a particular product rather than to optimize for what is best for patients).

It seems like 3 is a symptom of issue 1 - if there were national standards (that everyone stuck to) then it would be easier to get reliable mortality statistics.

It also seems that 3 highlights another issue - that many healthcare organizations that collect important data are not connected to hospitals, don't share that data with them, and that manifests in multiple issues. Perhaps this could be addressed in the recommendations?

The paper is not particularly patient centric. More could be included about the direct benefit to patients.

Although the need for standards and better clinical coding is mentioned, specifics are not discussed. Is there an alternative to ICD-10 / SNOMED-CT? What types of organizations are not using FHIR, and should they?

Who should these recommendations be followed by? Every organization that processes health data?

Reviewer #2 (Remarks to the Author):

The authors are making recommendations for achieving learning, interoperable and interconnected medical data systems. I agree these are laudable goals, and I think the authors' analysis of the problems is in many ways accurate. I am by no means convinced though, that their solution would bring us close to resolving these issues.

I have several important concerns. One is that the authors have given almost no credit to those who have done foundational work in standards development (though Dr. Hammond is a coauthor on this paper and is one of the fathers of standards). We have for example made great progress in the domain of content standards, and now have a single standard for all the main types of clinical data in the U.S. There is no mention though either of how this maps to standards used in other countries.

Another is the way they consider standards. Most of the criticism is focused on the "interoperability" standards.

That is not the usual terminology for standards, which are generally broken down into terminology, content, data exchange and privacy/security. This needs to be described more than it has been. If we want to get to interoperability, there are improvements which are needed in all four of these categories,

not just in data exchange. For example in content, although we have established standards for the main domains, each of those needs some additional building out at the margins.

The tables are not numbered.

Specific:

Line 133. I am not sure what the authors mean here; the EHRs are actually required to use data exchange standards such as HL7. It seems like what the authors are looking for is actually some type of conformance testing, which is required in the UK for example for some types of data.

Lines 142 and 143. Standard APIs are actually required now, and they are quite useful and used for many functions.

Line 181. The Makary reference is generally not believed by those in the safety community to be

accurate. Suggest citing either something else (like the To Err Is Human report or the JP James report--this latter may also be too high).

Line 241-2. Ironically, in health insurance exchanges lab exchange works by far the best. This seems like a case of the authors taking what they know and offering that as a panacea.

Line 280. Drug development is rounding error in the bigger issues of the costs of health care.

Line 294. But the UK did not succeed in the dexamethasone study because they had the items described in the preceding paragraph--it was largely because it had invested in the development of the Spine and widespread conformance testing, especially for laboratory data, which is not required in the U.S. Similarly, Israel's system succeeded to the degree it did largely because one company focused on developing data exchange as its business model (dBMotion).

P. 21. Bullet 2. Are the authors suggesting a new agency? That is likely to be a non-starter politically.

P. 21. General. Who is going to take all these actions? Is this just one big table? Overall, this needs some more work.

P. 22. Bullet 6. Who would set up these automated data verification processes?

Reviewer #3 (Remarks to the Author):

The manuscript gives an overview of the possible recommendations on the topic that, although not entirely original, are useful as a consolidated organic view on a possible approach to be taken.

There are some deficiencies that in view of the reviewer should be addressed:

1) the recommendations with regards to convincing patients to participate are based on rational arguments, that can be accepted on a rational level. However often refusal of these methodologies is based on emotional issues and mistrust of governments/big corporations. In recent years, this is all the more true. The paper does not mention involving patients as decision makers/consultative parties on the framework. This might sound paternalistic. They also should be partners of the proposed enforcement federal agency.

2) Ethical issues are not covered

3) Using only drop down lists may limit physicians freedom in keeping medical records. (page 23) To avoid duplicate records and an increased data entry burden, a physician-only (?) free text accessible area to record observations/comments could be foreseen

4) This brings us to the issue of who has access to the records, to the unique patient identifiers, and whether different access levels should be foreseen. Traceability of who has accessed what and whether they modified a record should also be foreseen.

The data storage provider should probably not be able to access personal identifiable information.

5) please expand on system failures/data backups/multi device compatibility.

6) line 248: clarify whether this relates to RWE or the Covid situation. It should not be the case in a well conducted RCT.

7) Federal agency: should not only map the roadblocks but identify the desirable characteristics of the best framework for best practices and standards. comment whether auditing by the agency will

also include suitability for GCP inspections.

Reviewer #4 (Remarks to the Author):

The concepts in this manuscript are well-defined and thoughtful; if implemented, they would transform the ways the US can learn from healthcare information in a radical and very positive way. This manuscript should most certainly be published and, hopefully, lead to further discussions and policy changes in our country. Following are my suggestions for minor revisions:

- a) The 'plain language' paragraph seems to need a lead-in sentence before jumping to the Amazon example (which does not appear until well into the manuscript).
- b) Line 126 - it seems there would be a better e.g. than imaging?
- c) "Requirements" of an LHS as a sub-header - should probably be 'a requirement' for and LHS? At this point, publications on LHS characteristics should be referenced (See the LHS Journal for several good references on this topic.)
- d) Also see the publication by Ros, Francisco et al. in the January 2021 LHS Journal. This deserves to be referenced since it addresses many of the same issues as in this manuscript - the need for a standards-based approach to healthcare data, especially for tracking Covid-19 information.
- e) This manuscript is obviously targeted towards the US, but references are made to other countries. Perhaps a paragraph explaining or acknowledging the US-centricity would be helpful?
- f) Another acknowledgement that may be warranted is that numerous public health systems still had to revert to paper during the pandemic and do not interface at all with EHRs.
- g) If I have understood this correctly, it seems that focusing on lab data initially (in the prototype?) may be inadequate and not necessarily representative of where key issues lie, e.g. with basic information such as demographics and possibly diagnosis, medical history and death data?
- h) Why not acknowledge CDASH as a global data collection standard for clinical research data? There are also mappings to HL7 standards done by CDISC and NCI. At least this represents tangible progress towards the stated goals. It seems that there was an effort not to name any standards or SDOs except ISO?
- i) This may be excessively 'politicized' to include in this manuscript, but another interoperability issue is the fact that it is nearly impossible for a person to access their own vaccine records if they are in databases separate from EHRs.
- j) A diagram/figure may be useful for readers, in addition to the table that was submitted with this manuscript.

Reviewer #5 (Remarks to the Author):

Data interoperability is a critical goal for health research, in order to be able to make use of the massive amounts of patient data that could be available, and this paper sets forth a clear overview of the main challenges for efforts to achieve data interoperability and recommendations for a path forward. While many of the points in the paper are not exactly novel, the paper is important for establishing a substantive and comprehensive set of recommendations to address this important problem.

One area that would strengthen the paper is doing more to acknowledge some larger context -- that data interoperability has been a longstanding issue, something that many have been calling for for years. What makes this moment/this set of recommendations different -- it would strengthen the paper if that were engaged more explicitly in the introduction. Also, in setting forth the challenges

for data interoperability, it could be useful to note which (considering the various efforts towards interoperability over the years) have proven the most challenging?

It also would be a point in favor of the author's arguments to note some of the health equity issues that are threaded through some of the items mentioned in the manuscript -- for example, they mention the need for the singular patient identifier; patients who are lower SES, who have no/less insurance coverage, tend to have more missing data in EHRs, tend to have to move more between different providers etc., meaning that they also may see less benefit from EHR-related health research; so the suggestion for the single patient identifier also has health equity ramifications in its favor. The argument regarding patient advisor engaged with the issue of privacy challenges, but in a rather limited way. More and more, data scientists have been pointing out the limitations of HIPAA for privacy protection, especially considering the increasing ease with which de-identified data may be re-identified. One thing to consider is that there are overall, there is a need to re-think privacy protections, moving more towards improved governance strategies, regulation to protect people from harms caused from data that can provide a more effective balance with the ability to use data for social/scientific benefit.

Finally, of the challenges stated, those involving business practices/proprietary software, seem to be the thorniest and have the least clear recommendations stated for presenting a path forward. It seems that should be acknowledged more as an issue - being more explicit about which measures are meant to take on that issue directly and how they do so (as well as why they seem feasible, despite this being a particularly challenging area to address).

Thank you.

Reviewer #1 (Remarks to the Author):

— Summary of paper —

This publication is a summary of recommendations to improve and modernize healthcare systems towards a world where an increasing range of software is used, where patients are increasingly taking control of their data and where sharing data for public health purposes with multiple public and private organizations is increasingly important.

At its heart the paper recommends a shift to standards for data collection. This is an important goal, and recommendations for healthcare organizations are helpful.

— Summary of review —

This is an important set of recommendations and I would recommend publication, but some points could be improved / expanded.

— Main comments —

There are a few points that the introduction could mention that these recommendations may help address:

The introduction does not mention some of the main emerging healthcare trends that necessitate solutions to the challenges the paper outlines, which I think is a missed opportunity. In 10-year's time, there will be many personal health records and apps (from EMR providers, tech companies, government agencies etc) that patients will use to store health data. If healthcare providers are not able to share data with these, patients may choose to go elsewhere. Patients often use apps to track their pregnancy or health, often sharing updates on social media. It is not hard to think that clinics which can interact with such apps will be more popular with patients - and given the public health benefits of PHRs (some governments internationally are promoting them), in years to come healthcare systems may be mandated to store patient generated data from wearables like Apple Watch / fitbit.

Reply: We agree with the reviewer and amended our paper to incorporate these concepts in the introduction.

There is also no mention of AI, which again relies on well curated datasets. As more organizations start to see benefits from AI in medical imaging, chatbots, etc, hospitals will miss out on the benefits without interoperability and data collection standards.

Reply: We are now mentioning these concepts in the Introduction.

Finally, I wonder if genomics will play a larger part in treatment, particularly in cancer treatment. Even at a local level, I think the ability to store this type of data is very poor, and certainly from a research perspective the scientific community will miss out on the opportunity for large scale GWAS studies without a better standard for data collection.

Reply: We totally agree with the reviewer, and amended our paper to incorporate these important concepts in the Introduction.

The paper outlines 5 major problems (no standards for data collection, no 'national PID', unreliable mortality data, poor clinical coding, EMR systems that are incentivized to lock a healthcare org into a particular product rather than to optimize for what is best for patients.

It seems like 3 is a symptom of issue 1 - if there were national standards (that everyone stuck to) then it would be easier to get reliable mortality statistics.

Reply: We have modified the organization of our discussion of major problems in our health information systems to reflect the reviewer's valid point.

It also seems that 3 highlights another issue - that many healthcare organizations that collect important data are not connected to hospitals, don't share that data with them, and that manifests in multiple issues. Perhaps this could be addressed in the recommendations?

Reply: We added this recommendation in Figure 1, as follows: *"Identify and link important health information data that are currently not connected to hospitals, including death, cause of death, data collected by registries, or vaccinations"*

The paper is not particularly patient centric. More could be included about the direct benefit to patients.

Reply: We agree that this is an important issue and we now point out repeatedly the benefits our proposed changes will bring to patients.

Although the need for standards and better clinical coding is mentioned, specifics are not discussed.

Reply: We have not discussed specific nomenclatures, because we didn't want them to be the focus of our paper. However, we added as a reference the following document that provides comprehensive information about the status of standards and clinical coding.

The Office of the National Coordinator for Health Information Technology. 2022 Interoperability Standards Advisory.

<https://www.healthit.gov/isa/sites/isa/files/inline-files/2022-ISA-Reference-Edition.pdf>

Note that this document assesses the health IT standards landscape *for data exchange*. It is for informational purposes only. It is non-binding and does not create nor confer any rights or obligations for or on any person or entity.

This seems to be the general rule in regulatory agencies. The only exception seems to be the with The National Highway Traffic Safety Administration's Office of Emergency Medical Services that provides a **universal standard** for the collection and transmission of emergency medical services (EMS) operations.

Is there an alternative to ICD-10 / SNOMED-CT?

Reply: The Convergent Medical Terminology (CMT) that Kaiser Permanente uses seems to be more practical and user friendly than **ICD-10 / SNOMED-CT** for medical personnel and patients.

What types of organizations are not using HL7 FHIR, and should they?

Reply: In the U.S.

1) those who don't really see a need to, since they are perfectly content with their current systems (that probably use HL7 V2 and especially CDA). We hear that billions of CDA documents are exchanged daily in the US alone). Not worth the trouble to them, .

2) those who see FHIR as a threat. This includes many of the big Health IT Vendors who have their own app stores based on their own proprietary APIs. Many of these vendors (including Epic and Cerner) also participate in the FHIR Community, but they still push their own solutions over FHIR.

3) those who just aren't aware enough to act. This includes many NIH researchers, but the NIH Office of Strategic Data Science which is putting together a major educational program **to increase awareness of the need to increase interconnectivity across systems**

The upcoming compliance dates at end of 2022 will push those in the first 2 categories, and the third group will engage over time. There are also countries (mostly in Europe) that are resisting use of FHIR because they have regulations in effect using other standards (often IHE with CDA) that they don't want to change. Interestingly, less developed, or relatively prosperous smaller countries in South America, Asia and Africa, are moving forward with FHIR, because it's easier to implement than older technologies.

The promise of all of these technological and scientific advances will be unrealized without interoperable standards that are fully representative of real-world data (not based on theoretical examples), and have been demonstrated to be fit-for-purpose for reliable data collection, exchange, integration, analysis, and full traceability to the original factual data.

Who should these recommendations be followed by? Every organization that processes health data?

Reply: We added this information in several parts of the paper, including in the plain language summary and in the Figure. For example, in the plain language summary, “Regulatory agencies providing reimbursements and every organization overseeing or processing health data should require fully tested, comprehensive, interoperable data collection and transmission standards to support automated integration and functions.

And in the figure: Address unwieldy data customization by enforcing interoperable and interconnected standards for medical data collection for every organization that collects or processes medical data (e.g., hospitals, laboratory information systems)

Reviewer #2 (Remarks to the Author):

The authors are making recommendations for achieving learning, interoperable and interconnected medical data systems. I agree these are laudable goals, and I think the authors' analysis of the problems is in many ways accurate. I am by no means convinced though, that their solution would bring us close to resolving these issues.

Reply: We agree that to resolve these “convoluted by design problems,” we need to go even deeper. Moving from exchange standards to data collection standards will, at a minimum, enable traceability to the original data facts. Traceability to the facts is critically important for troubleshooting systems and data failures and for building better realistic systems that can effectively address current problems.

I have several important concerns. One is that the authors have given almost no credit to those who have done foundational work in standards development (though Dr. Hammond is a coauthor on this paper and is one of the fathers of standards). We have for example made great progress in the domain of content standards, and now have a single standard for all the main types of clinical data in the U.S.

Reply: To address this concern, we included as a reference the very exhaustive assessment of the stage of the different data standards done by the Office of the National Coordinator*.

We agree that a lot of work has been done to develop the content standards for the main types of clinical data using HL7 *for data exchange*. However, we are still at a stage where we have only partially figured out the suitability of these standards for valid clinical data collection.

We have not discussed specific nomenclatures, because we didn't want them to be the focus of our paper. However, we added as a reference the following document that provides comprehensive information about the status of standards and clinical coding.

*The Office of the National Coordinator for Health Information Technology. 2022 Interoperability Standards Advisory.

<https://www.healthit.gov/isa/sites/isa/files/inline-files/2022-ISA-Reference-Edition.pdf>

This document is still for informational purposes only, and non-binding.

Of note: The National Highway Traffic Safety Administration's Office of Emergency Medical Services provides a ***universal standard*** for the collection and transmission of emergency medical services (EMS) operations.

There is no mention though either of how this maps to standards used in other countries.

Reply: We added within the **1. Lack of standardized data collection and transmission standards** section the following text:

“On the bright side, a recently initiated collaborative effort between HL7 International and OHDSI should provide a single common data model for sharing information in clinical care, and observational research. This effort, when expanded to data collection, can provide the building blocks for a comprehensive interoperable and interconnected HIS.²⁶ The Observational Medical Outcomes Partnership (OMOP) common data model (CDM) is being used in multiple countries to transform their data into the OMOP CDM.”

Another is the way they consider standards. Most of the criticism is focused on the "interoperability" standards. That is not the usual terminology for standards, which are generally broken down into terminology, content, data exchange and privacy/security. This needs to be described more than it has been. If we want to get to interoperability, there are improvements which are needed in all four of these categories, not just in data exchange. For example, in content, although we have established standards for the main domains, each of those needs some additional building out at the margins.

Reply: We agree that we have established standards for the main domains, and that each of those needs some additional building out at the margins. We have made great progress in the area of data exchange, enabling the creation of several good APIs. However, we have learned from artificial intelligence and machine learning that getting to the data facts *as they are generated* is critical for success in these areas. To have comprehensive data collection standards *based on real-data*, we have to actually investigate and identify which is the clinically relevant information that we need to capture. Only then when we will be able to access the factual data, discover the missing and erroneous pieces, and adjust the established data codes that do not fit perfectly well to capture clinically relevant data facts, we will be able to improve our understanding of clinical data, and built reliable decision support systems.

We agree that we need comprehensive, complete solutions that work across the continuum of health care, for the four categories listed above, and to enable the automated, comprehensive collection and use of all the needed information.

The tables are not numbered.

Reply: We have converted the table to a figure due to length limitations of the journal.

Specific:

Line 133. I am not sure what the authors mean here; the EHRs are actually required to use data exchange standards such as HL7. It seems like what the authors are looking for is

actually some type of conformance testing, which is required in the UK for example for some types of data.

Reply: Our concern is that vendors of certified EHRs that may be complying with exchanging data using HL7, are not necessarily using the same variables and values for use cases that are not covered by the certification rule, or internally across the multiple unique versions of their EHRs. Conformance testing that is required in the UK may be much more difficult to implement in the U.S., given the level of unique customization of our EHRs

We have added in the same section the following text: *“For example, EHRs modify most lab data received in their EHR databases by mapping them to the built-in terms in each implementation, creating roadblocks for data integration. This results in a multitude of data conversion cycles that are difficult to document and untangle, because they are not traceable back to the original data elements.”*

Lines 142 and 143. Standard APIs are actually required now, and they are quite useful and used for many functions.

Reply: There are ongoing and successful efforts to support data exchange via secure application programming interfaces.¹⁷ We have made great progress in the area of data exchange, enabling the creation of several good interfaces. “However, without a common, enforceable, and well-documented interoperable data structure and coding across pertinent health information systems and application programming interfaces, data exchange may still require a manual, frequently blinded mapping, making rapid transfer of needed information unfeasible.”

Line 181. The Makary reference is generally not believed by those in the safety community to be accurate. Suggest citing either something else (like the To Err Is Human report or the JP James report--this latter may also be too high).

Reply: Deleted the Makary reference and added the IOM reference

Line 241-2. Ironically, in health insurance exchanges lab exchange works by far the best. This seems like a case of the authors taking what they know and offering that as a panacea.

Reply: Although billing codes for lab data are rather standard, they are limited in scope. There are only 12 procedural CPT codes for billing for COVID-19 laboratory vs. 357 codes for Covid-19 tests authorized by the FDA. In addition, the health insurance data is typically unlinked from the full clinical data for a patient.

Line 280. Drug development is rounding error in the bigger issues of the costs of health care.

Reply: *“Adoption of our recommendations will simplify the continual enhancement, maintenance, oversight and the analytical functions of a fully functional, interoperable, public health and medical data system. The savings achieved through interoperability across the Research and Development value chain would expedite the discovery and development of safe and effective vaccines, and the identification of marketed drugs that can be repurposed to treat patients. Analysts will be able to discover consistent and reproducible efficacy and safety signals within and across multiple data resources and perform meta-analyses of all selected data rather than focusing on limited, static summary reports.”*^{36,37}

Line 294. But the UK did not succeed in the dexamethasone study because they had the items described in the preceding paragraph--it was largely because it had invested in the development of the Spine and widespread conformance testing, especially for laboratory data, which is not required in the U.S. Similarly, Israel's system succeeded to the degree it did largely because one company focused on developing data exchange as its business model (dBMotion).

Reply: With respect to pandemic response, countries that have more interconnected systems have been able to analyze their medical data more efficiently, and have provided important findings. For example, the rapid completion of the pivotal dexamethasone study in the United Kingdom in patients with COVID-19^{38,39} would have been very difficult to achieve in the U.S. with our highly customized, uncoordinated data systems for capturing patient level data. This study also benefited from the development of the Spine system in England that supports the IT infrastructure for health and social care joining together over 23,000 healthcare IT systems in 20,500 organizations and England’s widespread use of widespread conformance testing, especially for laboratory data. Also, Israel with its standardized, highly interoperable medical information system and a universal patient identifier, is providing critical information about breakthrough infections in patients who are fully vaccinated.^{40,41,42} Israel also benefited from the development and use of a data exchange system that can analyze non-centralized and non-standardized medical records to obtain needed public health information covering the broadest population possible.

P. 21. Bullet 2. Are the authors suggesting a new agency? That is likely to be a non-starter politically.

Reply: We recommend steps for action by specific bodies (in several bullets of Figure 1) being coordinated by the **Office of the National Coordinator for Health Information Technology** (lines 1-2 in the footnote of Figure 1)

P. 21. General. Who is going to take all these actions? Is this just one big table? Overall, this needs some more work.

Reply: Our list of recommendations (Figure 1) is now indicating in general terms who is responsible for taking the recommended actions.

P. 22. Bullet 6. Who would set up these automated data verification processes?

Reply: We added in Figure 1 that the Consortium has to establish the following: *Use automated data verification processes to confirm that the data collected are transmitted without distortion to correct patient records and end users, identify problems through feedback loops, and correct the sources of any data errors*

Reviewer #3 (Remarks to the Author):

The manuscript gives an overview of the possible recommendations on the topic that, although not entirely original, are useful as a consolidated organic view on a possible approach to be taken.

There are some deficiencies that in view of the reviewer should be addressed:

1) the recommendations with regards to convincing patients to participate are based on rational arguments, that can be accepted on a rational level. However often refusal of these methodologies is based on emotional issues and mistrust of governments/big corporations. In recent years, this is all the more true. The paper does not mention involving patients as decision makers/consultative parties on the framework. This might sound paternalistic. They also should be partners of the proposed enforcement federal agency.

Reply: we are now including patients as decision makers and we have added numerous references to the interests of patients in our MS:

Two of our recommendations now read as follows:

- *“Catalogue real-life clinical use cases needed for improving the quality of patient care, research, surveillance, and regulatory activities, and address current risks (medical errors due to a lack of patient identifiers, failure to transmit information to the right patient), and educate pertinent bodies and patients about these deficiencies”*
- *“Include patients in efforts to develop frameworks for interoperability and interconnectivity”*
- *“Require incentives aligned with patients’ and public health needs in which healthcare vendors are rewarded for documenting and avoiding medical errors and unnecessary processing costs.”*

2) Ethical issues are not covered

Reply: We added this important issue in several areas of our manuscript:

At the end of the plain language summary:

“Requiring comprehensive and universally binding standards ^{Error! Bookmark not defined.} rather than incentivizing voluntary and often piecemeal efforts for data exchange will simplify and speed up the generation of the valid, critically important analytical information environment that patients need.⁵

By the end of the introduction, we state the following:

“Recognizing our fiduciary ethical responsibility to rapidly provide the best the best information to help patients,⁵ we propose building an alternative, transparent system based on interoperability that starts at the data collection stage, to enable access to the factual data that can better support patient safety and scientific innovation while reducing costs. This alternative system would be one in which the benefits of new computational technologies can be realized, where patients are able to take control of their data, and where accurate and timely data can be rapidly shared to advance medical research and improve public health.”

3) Using only drop-down lists may limit physicians’ freedom in keeping medical records. (page 23) To avoid duplicate records and an increased data entry burden, a physician-only (?) free text accessible area to record observations/comments could be foreseen

Reply: We removed “drop-down” because it confuses the reader.

We now state:

“In those cases where automated data entry is not possible, an interoperable system should facilitate data entry and coding by providing automated, interactive graphic representations of the data already in the system and smarter options with standard terminology for outcomes for given symptoms, diseases, medications, and patient profiles.”

4) This brings us to the issue of who has access to the records, to the unique patient identifiers, and whether different access levels should be foreseen. Traceability of who has accessed what and whether they modified a record should also be foreseen.

Reply: *“The approach to governance we adopt must at a minimum ensure the following: The system is able to identify and control who can have the authorized level of access to the medical records. Every user has a unique Identification and a secure password. Audit trails are used to track every user activity, and to provide accountability. Only authorized personnel can access audit trails, and assess who has accessed or modified a record. And the data storage provider is not able to access personal identifiable information.”*

5) please expand on system failures/data backups/multi device compatibility.

Reply: We added within the section: 1. Lack of standardized data collection and transmission standards.

“To achieve a health system that enables continuous improvements, we need to collect the data that are most important for patient care, for accomplishing critically needed analytical

outcomes, for enhancing the level of evidence, and for potential use for addressing public health challenges.^{22,23,24,25} Therefore, we must focus on developing universal standards for the collection and validation of the most clinically important data as they are created (e.g., results from centrally calibrated laboratory tests during the entire course of clinical care, and the destination of intended users). Only when such standards are in place can we ensure that valid information is being correctly delivered. We must also ensure that the diverse software, transfer engines, and information technology systems can correctly interpret these standards, and process standard nomenclatures and notations without corruption. Redundant backup systems, feedback loops for prompt and early identification and communication of problems, and automated data verification processes will be needed to ensure data integrity and identify and correct the sources of transmission errors. Options should be provided for the public to monitor the accuracy of their medical data throughout all encounters (e.g., prescriptions, diagnosis, procedures), just as they check Social Security or bank statements.”

6) line 248: clarify whether this relates to RWE or the Covid situation. It should not be the case in a well conducted RCT.

Reply: This relates across the data in health information systems, including the data used to generate RWE and data in EHRs and Clinical trials (different Pharma companies select different Central Labs for their studies so the calibration of equivalent tests is done differently).

We state: “we suggest that one area that we can use as a model for how to achieve interoperability of patient data, where favorable incentives for reform may already exist, is in the processing of clinical laboratory data of drug marketing applications submitted to the FDA. Currently, such data undergo multiple transformation steps before regulatory submission, and although results in a given new drug application may be centrally calibrated, the results for many equivalent analytes coming from different sponsors, laboratories, instruments, and real-world data are not necessarily calibrated the same way.^{28,29,i”}

7) Federal agency: should not only map the roadblocks but identify the desirable characteristics of the best framework for best practices and standards. comment whether auditing by the agency will also include suitability for GCP inspections.

Reply: We agree that we need to build quality data systems and quality data inspections.

“To build quality data systems, we need to implement Good Clinical Practice inspections that identify and assess the application of best practices, including the usability, integrity, completeness, and traceability of the data, as well as reaching desirable health outcomes such as sound prediction algorithms, and the avoidance of preventable medical errors. These inspections need to monitor the adoption of interoperable data standards and the adherence and transmission of intact data to end users of HIS (as is being done by the Treasury

Department to assure honesty of monetary transactions). These inspections will require highly trained professional capable of detecting shadow data systems or other anomalies”

Reviewer #4 (Remarks to the Author):

The concepts in this manuscript are well-defined and thoughtful; if implemented, they would transform the ways the US can learn from healthcare information in a radical and very positive way. This manuscript should most certainly be published and, hopefully, lead to further discussions and policy changes in our country. Following are my suggestions for minor revisions:

a) The 'plain language' paragraph seems to need a lead-in sentence before jumping to the Amazon example (which does not appear until well into the manuscript).

Reply: Done (removed the Amazon example from the 'plain language summary' text)

b) Line 126 - it seems there would be a better e.g. than imaging?

Reply: we changed the example.

We are now stating: *“(e.g., results from centrally calibrated laboratory tests during the entire course of clinical care).”*

c) "Requirements" of an LHS as a sub-header - should probably be 'a requirement' for an LHS?

Reply: We joined this section with the previous one, and named it: **4. Business practices that hinder modernization and how to enable continuous improvements.**

At this point, publications on LHS characteristics should be referenced (See the LHS Journal for several good references on this topic.)

Reply: We added the following text and references: *“To achieve a health system that enables continuous improvements, we need to collect the data that are most important for patient care, for accomplishing critically needed analytical outcomes, for enhancing the level of evidence, and for potential use for addressing public health challenges.”*^{22.23.24.25}

d) Also see the publication by Ros, Francisco et al. in the January 2021 LHS Journal. This deserves to be referenced since it addresses many of the same issues as in this manuscript - the need for a standards-based approach to healthcare data, especially for tracking Covid-19 information.

Reply: Thank you for your suggestion. We added references related to learning systems, including the publication of Ros et al.

Ros et al argue that standardized data of the type now common in global regulated clinical research is the essential fuel that will power a global system for addressing (and preventing) current and future pandemics. We would add however that we cannot assume that the standardized data that are currently collected for clinical research contain the right level of detail to support multiple new uses. This is unlikely to be the case. For example, it is critically important to understand the timeline of the disease in individuals and in affected populations. As we emphasize in our paper, we need access to information that is fully traceable to the data facts information to ensure that it is correct.

e) This manuscript is obviously targeted towards the US, but references are made to other countries. Perhaps a paragraph explaining or acknowledging the US-centricity would be helpful?

Reply: We have modified the Introduction so, that it now reads as follows: Healthcare continues to be one of the most fragmented sectors of the U.S. economy with countless non-interoperable and non-interconnected data systems.²

We added: *“The following description of deficiencies in the U.S. health information system and the accompanying recommendations for addressing them at their root causes are meant to stimulate fruitful dialogues among multiple stakeholders and inform policy changes in the US and other countries where such measures are needed.”*

f) Another acknowledgement that may be warranted is that numerous public health systems still had to revert to paper during the pandemic and do not interface at all with EHRs.

Reply: We added in the introduction: *“During the pandemic numerous public health systems and practitioners were forced to continue to use outmoded forms of communication (e.g., paper and fax) which do not adequately provide rapid access to needed information.”*

g) If I have understood this correctly, it seems that focusing on lab data initially (in the prototype?) may be inadequate and not necessarily representative of where key issues lie, with basic information such as demographics and possibly diagnosis, medical history, and death data?

Reply: Please refer to the **recent** three publications that we listed in our amended manuscript describing the problems resulting from the inadequate or absence of reagents to calibrate equivalent tests done by different instruments and laboratories.

We also included in the text referring to our proposed prototype the following: *“To ensure that the data are analyzable, this initiative will also make use of all the most advanced standardized data elements that are necessary for the analysis of each regulatory data submission, including those related to demographics, diagnosis, medical history, death, and cause of death.”*

h) Why not acknowledge CDASH as a global data collection standard for clinical research data? There are also mappings to HL7 standards done by CDISC and NCI. At least this represents tangible progress towards the stated goals. It seems that there was an effort not to name any standards or SDOs except ISO?

Reply: We have not discussed specific nomenclatures, because we didn’t want them to be the focus of our paper. However, we added as a reference the following document that provides comprehensive information about the status of standards and clinical coding (including CDASH).

The Office of the National Coordinator for Health Information Technology. 2022 Interoperability Standards Advisory.

<https://www.healthit.gov/isa/sites/isa/files/inline-files/2022-ISA-Reference-Edition.pdf>

Accessed on January 11, 2022

This document is still for informational purposes only, and non-binding.

The National Highway Traffic Safety Administration’s Office of Emergency Medical Services provides a **universal standard** for the collection and transmission of emergency medical services (EMS) operations.

CDASH is being assessed in the *The Office of the National Coordinator for Health Information Technology. 2022 Interoperability Standards Advisory* document.

One of the CDASH goals is to establish a standard nomenclature for data entered in Case Report Forms of a Clinical Trial. CDASH is not in the FDA catalog, probably because CDASH is a data collection standard for Case Report Forms that FDA does not regulate, although of course FDA requires annotated CRFs be submitted and that may have CDASH tags on it.

Moreover, progress has not yet reached the level needed for enabling the automated integration of such data and their use to support automated, fully traceable analytical functions. We are just learning which are the data elements that CDASH needs to capture for a variety of different diseases, understanding how they match with the multiple versions of HIS and how to manage and properly report (i.e., in a publication) the matching gaps in the HIS and in CDASH and how they were handled in each situation.

i) This may be excessively 'politicized' to include in this manuscript, but another interoperability issue is the fact that it is nearly impossible for a person to access their own vaccine records if they are in databases separate from EHRs.

Reply: We added: *“(e.g., it is still nearly impossible for a person to access their own vaccine records if they are in databases separate from their EHR records or were submitted by paper or fax.)”*

j) A diagram/figure may be useful for readers, in addition to the table that was submitted with this manuscript.

Reply: We replaced the table with a Figure

Reviewer #5 (Remarks to the Author):

Data interoperability is a critical goal for health research, in order to be able to make use of the massive amounts of patient data that could be available, and this paper sets forth a clear overview of the main challenges for efforts to achieve data interoperability and recommendations for a path forward. While many of the points in the paper are not exactly novel, the paper is important for establishing a substantive and comprehensive set of recommendations to address this important problem.

One area that would strengthen the paper is doing more to acknowledge some larger context -- that data interoperability has been a longstanding issue, something that many have been calling for years. What makes this moment/this set of recommendations different -- it would strengthen the paper if that were engaged more explicitly in the introduction.

Reply: Added by the end of the introduction: *“Data interoperability is a goal that many parties have been long been advocating. What makes this issue more urgent than before is our realization that we have ‘to move faster than the virus’ [Personal communication from Dr. Mirta Roses], (at a time when hospitals are being saturated with more patients that they can handle), and that current information solutions, which are focused on non-binding interoperability standards for data exchange,¹ or on multiple, unreconciled Common Data Models hamper our ability to act quickly.”*

Also, in setting forth the challenges for data interoperability, it could be useful to note which (considering the various efforts towards interoperability over the years) have proven the most challenging?

Reply: Added to the introduction the following: *“Perhaps the most challenging roadblock for implementing universal binding interoperability for data collection is our hesitancy to change particular business, software, and data management practices that generate reliable revenue. These practices inevitably lead to reliance upon incomplete, unsynchronized coding and mapping solutions, and poorly defined communication paths for improving decision making by institutions, vendors of HIS, software developers, and stakeholders (including patient advocates).*

“Recognizing our fiduciary ethical responsibility to rapidly provide the best the best information to help patients,⁵ we propose building an alternative, transparent system based on interoperability that starts at the data collection stage, to enable access to the factual data that can better support patient safety and scientific innovation while reducing costs

It also would be a point in favor of the author's arguments to note some of the health equity issues that are threaded through some of the items mentioned in the manuscript -- for

example, they mention the need for the singular patient identifier; patients who are lower SES, who have no/less insurance coverage, tend to have more missing data in EHRs, tend to have to move more between different providers etc., meaning that they also may see less benefit from EHR-related health research; so the suggestion for the single patient identifier also has health equity ramifications in its favor. The argument regarding patient advisor engaged with the issue of privacy challenges, but in a rather limited way.

Reply: We now state: *“A single patient identifier has health equity ramifications in its favor. Patients who are poorer typically have less insurance coverage or none at all and often change health care systems. They are underrepresented in health information systems and research studies, and less likely to have their specific needs understood. A unique identifier should improve the representation of these patients in our HIS and thus our ability to address health inequities.”*

More and more, data scientists have been pointing out the limitations of HIPAA for privacy protection, especially considering the increasing ease with which de-identified data may be re-identified. One thing to consider is that there are overall, there is a need to re-think privacy protections, moving more towards improved governance strategies, regulation to protect people from harms caused from data that can provide a more effective balance with the ability to use data for social/scientific benefit.

Reply: *“As data scientists, we are concerned about the limitations of HIPAA for privacy protection, due to the ease that such data can be re-identified. Our laws and regulations need to balance individual privacy protection, with making data available for improving health outcomes. At a minimum, the approach to governance we adopt must ensure the following: The system is able to identify and control who can have the authorized level of access to the medical records. Every user has a unique ID and a secure password. Audit trails are used to track every user activity, and to provide accountability. Only authorized personnel can access audit trails, and assess who has accessed or modified a record. And the data storage provider is not able to access personal identifiable information.”*

Finally, of the challenges stated, those involving business practices/proprietary software, seem to be the thorniest and have the least clear recommendations stated for presenting a path forward. It seems that should be acknowledged more as an issue - being more explicit about which measures are meant to take on that issue directly and how they do so (as well as why they seem feasible, despite this being a particularly challenging area to address).

Reply: Added the following text: *“Perhaps the most challenging roadblock for implementing universal binding interoperability for data collection is our hesitancy to change particular business, software, and data management practices that generate reliable revenue. These*

practices inevitably lead to reliance upon incomplete, unsynchronized coding and mapping solutions, and poorly defined communication paths for improving decision making by institutions, vendors of HIS, software developers, and stakeholders (including patient advocates).

Recognizing our fiduciary ethical responsibility to rapidly provide the best the best information to help patients, ^{Error! Bookmark not defined.} we propose building an alternative, transparent system based on interoperability that starts at the data collection stage, to enable access to the factual data that can better support patient safety and scientific innovation while reducing costs.

Thank you.

Reply: Thank you!

REVIEWERS' COMMENTS:

Reviewer #1 (Remarks to the Author):

The authors have accepted my comments and revised their manuscript accordingly. I am happy to recommend the article for acceptance.

Reviewer #2 (Remarks to the Author):

The authors have addressed all my major issues in satisfactory ways.

Reviewer #3 (Remarks to the Author):

manuscript acceptable with revisions as done

Reviewer #4 (Remarks to the Author):

Very nice job addressing comments from all of the reviewers!

Reviewer #5 (Remarks to the Author):

Looking over the authors' responses and changes to the manuscript, the authors' attention to responding to the range of concerns is very appreciated - they have integrated appropriate responses, including to the ethical issues - thank you!